# Would the United States Have Had Too Few Beds for Universal Emergency Care in the Event of a More Widespread Covid-19 Epidemic?

**DOI:** 10.3390/ijerph17145210

**Published:** 2020-07-19

**Authors:** Rodney P Jones

**Affiliations:** Healthcare Analysis & Forecasting, Wantage, Oxfordshire OX12 0NE, UK; hcaf_rod@yahoo.co.uk

**Keywords:** hospital bed numbers, Covid-19, excess deaths, population density, capacity pressures, international comparison, methods

## Abstract

(1) Background: To evaluate the level of hospital bed numbers in U.S. states relative to other countries using a new method for evaluating bed numbers, and to determine if this is sufficient for universal health care during a major Covid-19 epidemic in all states (2) Methods: Hospital bed numbers in each state were compared using a new international comparison methodology. Covid-19 deaths per 100 hospital beds were used as a proxy for bed capacity pressures. (3) Results: Hospital bed numbers show large variation between U.S. states and half of the states have equivalent beds to those in developing countries. Relatively low population density in over half of US states appeared to have limited the spread of Covid-19 thus averting a potential major hospital capacity crisis. (4) Conclusions: Many U.S. states had too few beds to cope with a major Covid-19 epidemic, but this was averted by low population density in many states, which seemed to limit the spread of the virus.

## 1. Introduction

The Covid-19 pandemic has caught health systems around the world by surprise. Covid-19 places a special burden on acute medical and intensive care beds [1,2,3]. The world’s largest economy, the USA, was also struggling to find the necessary resources to battle the virus. As with many countries, supplies of protective equipment and ventilators were initially in short supply [4].

New York experienced an exceptionally large number of Covid-19 deaths [5]. Are there any lessons to be learned from the outbreak so far within the context of a healthcare system in which both primary and secondary care are mainly resourced to deal with just insured patients [6]?

This study will seek to evaluate if U.S. states had enough hospital beds to cope with a large epidemic across all parts of the country and will use a new method for comparing inter- and intra-national bed numbers to do so.

For more than four decades economists have known that health care costs and resource usage escalate toward the end of life, more so than from age per se [7,8,9]. This is especially so for hospital inpatient care with numerous studies demonstrating increased admissions and bed occupancy toward the end of life, which is moderately independent of the age at death [10,11,12,13].

This has collectively been called the nearness to death (NTD) effect or the time to death (TTD) effect. One study indicated that while 45% of lifetime admissions occurred in the last year of life some 55% of bed occupancy occurred in this period [14].

Unfortunately, these relationships have never been incorporated into methods for forecasting healthcare demand and capacity planning. Such methods almost exclusively rely on time series analysis or variants of demographic or age-based forecasting [14,15,16,17,18,19]. The limitations of these methods are discussed elsewhere [15,18,19,20,21,22,23].

Research into better methods for comparing hospital bed numbers between countries and regions has culminated in a new method which relies on the operation of the NTD effect [24,25]. Based on this observation, bed numbers in different countries can be plotted on a graph showing beds per 1000 deaths (Y-axis) versus the log of deaths per 1000 population (X-axis), also called the crude death rate. The deaths per 1000 population serve as a measure of age structure while beds per 1000 deaths reflects the role of nearness to death.

This method avoids the limitations of simplistic international comparisons based on beds per 1000 population [20], which contains no adjustment for age structure or nearness to death. Regarding the use of deaths per thousand population it should be noted that as far back as 1981 a relationship between bronchitis and emphysema (men aged 65–74) admissions per 1000 population and deaths per 1000 population had been demonstrated [19].

In this new method a set of parallel lines (by changing the intercept) can be drawn which intersect the bed provision in different countries.

The average for developed countries has the equation [25]:Beds per 1000 deaths = 1057 − 230.3 × ln(deaths per 1000 population),(1)

The country with the highest intercept is Japan; however, this arises from ambiguity in the definition of a hospital bed where Japan counts the equivalent to nursing home care in its definition of a hospital bed [25]. The lowest bed availability in the developed world, where comprehensive health care is provided for the *entire* population, can be found in Singapore and New Zealand. This has been achieved by over 20 years of government policy and policy implementation in integrated care [24,25]. For these two countries the intercept in the above equation is 790 rather than 1057 for the international average.

The method was validated at sub-national level using beds in the states of Australia where a long-standing deficit in bed numbers in the state of Tasmania was correctly quantified [25]. The method has subsequently been confirmed to apply at local level using data for the populations covered by Clinical Commissioning Groups in England (submitted).

Total hospital bed numbers have been used in this study because they are routinely available [26]. The number of medical beds is not generally available except for several European countries and only up to the year 2009 [27].

The health insurance system and population distribution in the U.S. places a set of constraints on physician and bed availability as resources follow a mix of population density and wealth, and not necessarily need [28,29,30]. These issues are addressed further in the discussion section.

## 2. Materials and Methods

### 2.1. Data Sources

In this analysis confirmed Covid-19 deaths were from Bing.com [5], total hospital bed numbers include adult and pediatric acute care, plus maternity and mental health. International total hospital beds were from the World Bank [26], total hospital beds by US state was from the Kaiser Family Foundation [31], while the number of acute beds by US state was from the American Hospital Directory [32]. Available beds are beds staffed to receive patients. State population and median household income was from United States Census Bureau [33,34]. Proportion rural population and rural poverty were from the US Department of Agriculture [35]. Land use was from the National Park Service [36]. Deaths in U.S. states was from the Centers for Diseases Control and Prevention (CDC) Wonder database [37]. Weighted population density was from Decision Science News [38]. All data is publicly available but can be obtained on request from the author.

### 2.2. Data Manipulation

All data was manipulated, and charts prepared using Microsoft Excel. Covid-19 deaths per bed was calculated by dividing deaths (at the 9th of June) by total hospital bed number as in the data sources. Adjusted bed number by state was calculated as the higher of two methods as follows. Total hospital bed numbers [31] were compared to acute bed numbers [32]. The former is from a non-mandatory survey while the latter are mandatory for Medicare. The ratio of one to the other was calculated for each state. The median value of this ratio was 12% higher for total hospital beds. In 28 states, total hospital beds were lower than acute beds and in those states the higher of total hospital beds or acute beds times 1.12 was used in this study. Beds in some mental health hospitals or smaller community hospitals may have been missing from the count of total beds.

All other numbers are derived by simple division of one parameter by the other, i.e., Covid-19 deaths per 100 total hospital beds.

International data in Figure 1 was from previous studies on this topic [24,25] while the parameter adjusted beds per 1000 deaths is detailed in previous studies [24,25].

## 3. Results

### 3.1. Beds per 1000 Population and Beds per 1000 Deaths Compared

Beds per 1000 population has been used to compare international bed numbers for many years and hence it is useful to see how this compares with beds per 1000 deaths. Such a comparison is given for US states in Figure 1 where it can be seen that, as expected, there is a degree of relationship between the two, such that, Beds per 1000 deaths is approximately equal to 109 × Beds per 1000 population. Typically, younger populations lie above the trend line while older ones lie below.

Applying the same approach to international data [24] gives a trend line with a slope of 121 as the international average (data not shown). However, a series of lines describe the international data with a slope ranging from 63 (mainly Russia and former Soviet countries and also West Virginia in Figure 1) to 172 (encompassing most countries), through to 600 (Kuwait, Bahrain which have the youngest populations and the least deaths per 1000 population).

While beds per 1000 deaths is a useful comparator the new method has the advantage that it uses both deaths and population to compare bed provision (next section).

### 3.2. Hospital Bed Numbers Compared

Figure 2 shows a plot using data for U.S. states in 2018 using the new method which also includes lines for the international average for developed countries, and for New Zealand and Singapore. Data for Canada (red diamond) and the UK (purple triangle) are also shown for comparison as examples of low bed number countries offering universal health care. Canada and the UK lie on a line parallel to but above that for New Zealand and Singapore. The data point for the entire USA is the red square. All US states, except for the federal district of Washington D.C. (District of Columbia) and to a less extent South and North Dakota, have far fewer hospital beds that the average for developed countries, while several states (Washington, Oregon) have less beds than the average for the less developed countries. Around half of U.S. states are below the most bed efficient line for universal care in Singapore and New Zealand.

For comparison, the international range in deaths per 1000 population is from around 1.0 in the oil rich gulf countries through to 17 in Sierra Leone. The UK has a ratio of 9.0 [18]. Figure 1 also includes a line describing the average for the less developed countries:Beds per 1000 deaths = 648 − 200 × ln(deaths per 1000 population),(2)

Countries near to the less developed countries average are Colombia, Ecuador, and Algeria. Around 20% of US states lie close to this line. Despite the lower slope for the line describing the less developed countries (Equation (2) above), over the limited range in deaths per 1000 population seen in US states (5.9 in Utah to 13.1 in West Virginia), the two equations are almost parallel and the equation for developed countries can be applied to adjust all countries to the US average of deaths per 1000 population (next section).

The average line for the least developed countries has the equation (analysis not shown):Beds per 1000 deaths = 318 − 111 × ln(deaths per 1000 population),(3)

The line for the least developed countries lies below the scale in Figure 2 and includes countries such as Sierra Leone, Madagascar, Senegal, Niger, etc.

### 3.3. US States with the Least Beds

By adjusting all U.S. states to the average deaths per 1000 population for the U.S., namely 8.69, and adjusting world countries to the same number allows direct comparison between U.S. states and their nearest international equivalent.

This comparison is given in Table 1 for the states with the least number of beds which are at or below the U.S. average. As can be seen Oregon has equivalent beds to Burundi, Washington (not to be confused with Washington D.C.) to Costa Rica, California has equivalent to Albania, and Colorado has equivalent to Algeria. The average for the USA is equivalent to Vanuatu.

Such low levels of bed provision clearly indicate that large parts of the U.S. may be susceptible to a major epidemic and specific issues relating to Covid-19 will now be investigated.

### 3.4. Population Density

The possibility that Covid-19 transmission may have been reduced in those states with lower population density is investigated in Figure 3.

Figure 3 uses county-level weighted population density [38] to adjust for the fact that the surface area of most states is largely devoted to agriculture, forestry, or national parks. For example, there are 345,000 km^2^ of national parks [36] which represent 1.4 times the size of the United Kingdom. Most of the population therefore experience a far higher (weighted) population density than the simple division of population by land area.

In Figure 3, Covid-19 confirmed cumulative deaths were up to the 9th June 2020 [5], while deaths in 2018 [37] (the most recent data available for the U.S.) were taken as the baseline per annum deaths in the absence of Covid-19. Hence excess Covid-19 deaths as a percentage of total deaths.

As can be seen in Figure 3 low population density is generally associated with lower excess Covid-19 confirmed mortality. Mortality generally rises for state weighted population density above 1000 persons per mile^2^ (390 persons per km^2^).

### 3.5. Covid-19 Deaths per Bed

Virus transmission in an epidemic always leads to a granular spatial distribution of cases [39]. The major focus of the Covid-19 epidemic in the U.S. was the heavily populated North East corner, with New York receiving the fifth highest number of international visitors in the U.S. [40].

In the absence of international hospital admissions due to Covid-19 the cumulative deaths per hospital bed can be used as a measure of potential capacity pressure upon available beds.

Figure 4 shows the relationship between cumulative deaths per bed and cumulative excess mortality for US states. As can be seen 36 states had less than 10 Covid-19 deaths per 100 beds (10%) over the 70-day period 1 March to 9 June which reflects the lower Covid-19 cases and deaths in those states with lower populations density. While New York received international attention due to the high number of deaths Figure 4 shows that New Jersey, Massachusetts, and Connecticut all experienced higher potential bed demand while Rhode Island experienced intermediate potential demand. Washington D.C. experienced relatively lower bed demand due to the high number of hospital beds in this Federal District.

The relatively low population density in most US states seemingly averted a catastrophic Covid-19 bed capacity overload.

## 4. Discussion

### 4.1. Healthcare in the U.S.

While many countries seek to develop a coordinated universal healthcare system, healthcare in the US is largely left to market forces [41]. Medicaid and Medicare provide some basic care to the poor and elderly, government workers and the military are covered by state and federal government schemes while the rest of the population is covered by private insurance via their employer or themselves, or to remain uninsured. Medicaid and Medicare recipients can take out additional insurance to complement the basic government scheme. [41,42]. There are over 900 medical insurers offering a complex variety of schemes with different terms and conditions [42].

Hospitals can be either for-profit or not-for-profit and are either independent, state owned or part of larger chains, and are free to make their own planning decisions regarding bed numbers. Some 14% of the US population lives in a rural area (range 0% Washington D.C., New Jersey, Delaware to 69% in Wyoming), 16.1% of the rural population lives below the poverty threshold (range 6.9% in Connecticut to 26.9% in Arizona) [35]. Rural populations in the US are known to access hospital services less than their urban counterparts due to distance and poverty [43]. However, even in the densely-populated UK with free access to healthcare, rural populations have 20% lower hospital bed occupancy than their urban counterparts [44].

The rural population is served by a multitude of small community hospitals, such that half of US hospitals have fewer than 100 beds [30]. In 2019 that US average was 191 acute beds per hospital (range 90 in Wyoming to 259 in New Jersey and Connecticut and 306 in New York) [32]. The US average occupancy rate for acute hospital beds is 55% (range 25% in Wyoming with 69% rural population to 76% in Washington D.C. with 0% rural population) [32]. The smallest hospitals tend to occur in the states with the lowest average income [30]. The small size of U.S. hospitals imposes low average occupancy [45], and higher unit costs [46], leading to hidden barriers to access acute care.

Hence healthcare in the U.S. is what many may regard as overly complex and with little government intervention to ensure universal health care. Generally poorer rural populations seek less acute care, which is then compounded by low population density leading to small hospital size with resulting marginal commercial viability. Hence the generally low bed numbers in Figure 1 and Figure 2 and Table 1 and the difficulty in explaining exactly which factors contribute to low bed numbers in some states.

### 4.2. Beds in the U.S.

Figure 1 has demonstrated that while beds per 1000 deaths and beds per 1000 population give roughly similar measures of bed availability, the new method gives greater insight into the issue of bed numbers since it includes both deaths and population as the drivers of bed demand.

As demonstrated in Figure 2 around 40% of U.S. states fall below the line for New Zealand and Singapore. States lying below the line for New Zealand and Singapore has bed availability levels equivalent to the “less developed” countries where universal health care is also not available [24,25]. Lowest bed availability occurs in Oregon. The Netherlands and Australia have total beds near the international average. Australia has a higher proportion of rural population than the U.S. [47], and rurality per se cannot be an explanatory factor for the low number of U.S. hospital beds.

Washington D.C. which has some of the most prominent hospitals and research institutes, has the highest number of physicians per capita, the highest median household income, and only 3.7% of persons are uninsured [34,48]. Washington D.C. is probably the closest approximation to universal healthcare in the U.S.

Countries like Germany and Austria which have 3-times the U.S. average number of beds [24] can manage a surge event such as Covid-19 with far greater ease [49].

It is well recognized that the U.S. healthcare industry only represents a proportion of the population and hence comparative International health outcomes are poor [3,50]. This is confirmed by the low levels of hospital bed provision in Figure 1 and Figure 2 and Table 1.

### 4.3. Medical Beds

While data on the number of medical beds in the US and elsewhere is not available it is obvious that the number of these beds must be lower than the total beds. For example, in England available mental health beds account for 15% of total hospital beds while maternity accounts for a further 6% [51], the medical group of specialties accounts for 55% of total *occupied* beds [52].

Research suggests that the demand for medical beds is directly proportional to the number of total deaths and has a far flatter slope than Figure 1 [53]. The average number of ***available*** (staffed) medical beds per 1000 deaths across Europe was 205 [27], with a 20-year average of 165 ± 5 (±standard deviation) ***occupied*** medical beds per 1000 deaths in England [53]. In England medical beds operate at over 95% average occupancy [51].

Some US states have barely enough **total** beds to accommodate just the medical group of patients under universal health care.

As in other countries, the US did shut down routine surgery to create space for Covid-19 patients [54]; however, the basic position of too few beds remains an overriding problem.

### 4.4. Critical Care Beds

The major impact of Covid-19 has been the high use of critical care beds [1,3]. As with medical beds, research shows that critical care bed requirements are likewise directly proportional to the number of deaths [55]. The U.S. has the highest number of critical care beds per population in the world [56] but did need additional critical care beds to service Covid-19 hot spots.

Under normal circumstances the poorer communities (not covered by Medicaid) affected by Covid-19 would not have been able to access hospital care [57,58], and many with health insurance, especially the elderly, would not seek medical care due to the risk of bankruptcy due to co-payments [57,58,59,60]. This would have acted to further spread the disease.

On the 3 April President Trump announced a Covid-19 support package in which the inpatient care of uninsured persons suffering from Covid-19 would be covered by the Federal government [61]. Some Insurers have also announced that co-payments would be waived. However, it is unclear how effectively this has been communicated and if hidden barriers to access remained such as access to Covid-19 testing and fear of a high bill for attending the emergency department should the illness not be due to Covid-19. Indeed, those recently unemployed appear to fall into a policy gap [58].

### 4.5. Specifics to Covid-19

A recent study has used age-specific mortality patterns along with population demographic data to map projected burden of COVID-19 and the associated cumulative burden on the healthcare system (total hospital beds and Intensive care beds) at county level in the US for a scenario in which 20% of the population of each county acquired infection. It was suggested that per capita disease burden and relative healthcare system demand may be highest away from major population centers where there are the fewest hospital and critical care beds [62].

The states identified in this study roughly align with those identified by Miller et al. [62] simply because too few beds, mainly located in large cities, are an inescapable limitation to treating Covid-19 patients requiring hospitalization. Figure 2 and Figure 3 shows the resulting granular distribution of Covid-19 confirmed deaths. States with extremely low bed provision such as Oregon were spared from a potential medical disaster by virtue of low incidence of cases relative to available beds.

### 4.6. Population Density

The U.S. has a somewhat unique population distribution. Only five cities (including New York at 33,000 per square mile) have a weighted population density above 10,000 persons per square mile [29,38,63,64]. By comparison London has a weighted density of 22,000 per square mile and some 40% of the population in England live at greater than 10,000 persons per square mile [29]. Much of the remainder of U.S. has low population density. Population density is a critical factor in disease and Covid-19 transmission [65,66]. This association for Covid-19 has been observed in English local authorities [67], and in England and Wales the age-standardized Covid-19 mortality rate in urban major conurbations was 6-times higher than in rural hamlets [68].

Figure 3 showed the relationship between the ratio of confirmed Covid-19 deaths (at 19 April 2020) per 1000 population versus the weighted population density for each state. Weighted population density is the effective population density experienced by the average person in the state [38,64]. Weighted population density considers that most people live in cities and towns hence the population density that they experience is much higher than the simple division of population by total land area. Hence this is between 12 to 60 times higher than the raw density [38,64]. Population density is not the only factor responsible for spread of Covid-19 and factors such as international exposure, household crowding and mass gatherings will also play a role [65,66]. While the extent of Covid-19 testing varies between US states, Figure 3 nevertheless establishes an important precedent.

As demonstrated the states with lowest weighted population density all experience low death rates. These states tend to have fewer hospitals simply because a hospital needs a high population within a reasonable travel distance to be financially viable. Hence low weighted population density in over half the states has acted to avert a medical disaster in the USA.

### 4.7. Why Such Low Bed Numbers in Some States?

Given the complexities of healthcare in the U.S. discussed in Section 4.1 the issue of exactly why some states have such low bed numbers has no easy answer. One study has demonstrated that the average number of beds per hospital was related to average state income [30]. Average length of stay in each state was related to average number of beds per hospital, being generally lowest in those states with the smallest hospitals, and lowest average income [30]. Another study demonstrated that the number of bed days per death (or occupied beds per 1000 deaths) rose with population density [29]. Population density and proportion rural population are interrelated, and it would therefore seem that urban populations seem to make greater use of hospital care at the end of life.

In past decades the U.S. was able to claim that lower length of stay (LOS) was an explanation for fewer beds; however, in recent years LOS in many countries is now reasonably close to or lower than that in the U.S. [69,70] and this can be dismissed as an explanatory factor.

Hence the reason why Oregon, a mainly agricultural state focusing on horticulture (16% rural, 15% rural poverty) has so few beds is a complex mix of numerous factors which seem not to have been adequately investigated, and which are largely irrelevant in the context of the U.S. health care industry. The proportion of persons who are insured is probably an overly simplistic indicator since insurance policies issued by 900 insurers can have widely different terms and conditions.

A simple overview is that in the U.S. healthcare resources follow “money” rather than “need” and this is then distorted by population density, or more correctly population distribution.

### 4.8. Limitations of the Study

Due to the absence of numbers of persons admitted to hospital for Covid-19 the total confirmed Covid-19 deaths has been used as a proxy measure. Clearly persons die of Covid-19 in nursing homes [71] and at home; however, this is probably a reasonable proxy for hospital capacity pressure since death is most likely to occur in older people who are most represented in Covid-19 admissions [72,73]. In New York it was recognized that elderly Covid-19 patients were overly rapidly discharged into nursing homes, which led to cross-infection and further compounded the number of deaths [74]. The same hasty discharge into nursing homes occurred around the world [71]. Covid-19 deaths are themselves an underestimate due to variable testing capacity between states and the real excess mortality may be 28% higher [75]; however, the available data must be used to make comparisons despite its limitations.

The second limitation is the use of total hospital beds to compare between states and countries. Once again given the lack of detailed international data covering just acute beds this is a reasonable proxy.

### 4.9. Further Research

The issue of international hospital bed number comparisons is highly topical and the new method for bed comparison is an improvement on the former simplistic beds per 1000 population. The new method should be used to delve further into the reasons why different healthcare systems operate with such widely varying bed numbers. Factors such as population density, average hospital size, and rurality need to be given greater prominence.

## 5. Conclusions

A new method for comparing international bed numbers has been used to give greater insight into the variation in bed numbers between U.S. states. Decades of focus on providing medical resources to the insured, along with high levels of co-payment imposed by insurers to discourage use of those resources [57,58,59,60] has left the US with very a very wide range in bed provision between states and generally low levels of hospital beds in relation to other developed countries. Some states have numbers of beds equivalent to those in less developed countries.

Despite extremely low bed numbers in many states the U.S. was spared from a full-scale disaster due to concentration of cases in a few states with relatively high bed numbers.

The healthcare “system” in the US is in a seemingly intractable dilemma from which it is difficult to extract itself and which leaves it open to the impact of future pandemics should they occur with a different spatial distribution or mode of transmission to Covid-19. Indeed, statistics indicate that spread of Covid-19 has increased in recent days [76].

## Figures and Tables

**Figure 1 ijerph-17-05210-f001:**
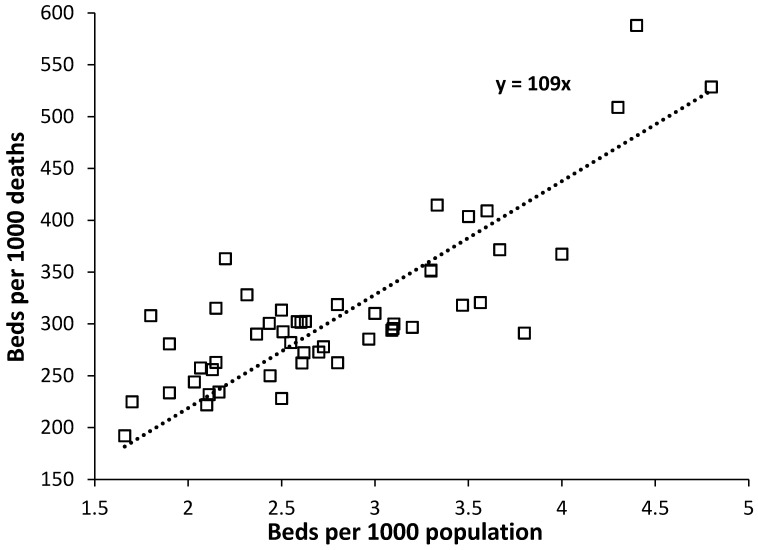
A comparison of U.S. states for beds per 1000 deaths versus beds per 1000 population.

**Figure 2 ijerph-17-05210-f002:**
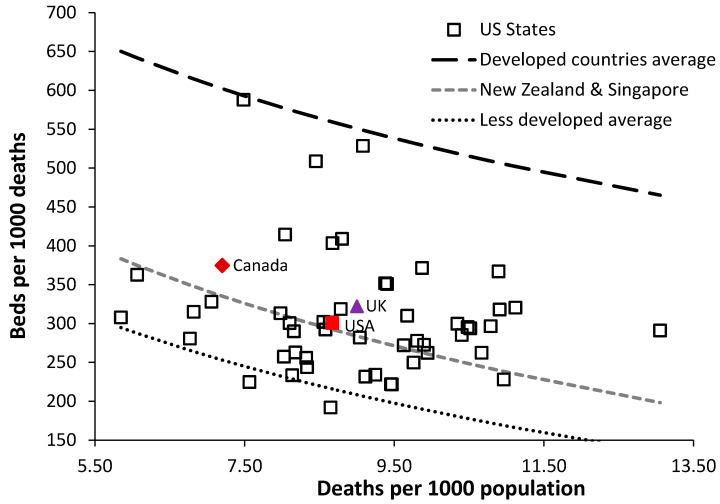
Details of total hospital bed availability in U.S. states in 2018 compared to international norms.

**Figure 3 ijerph-17-05210-f003:**
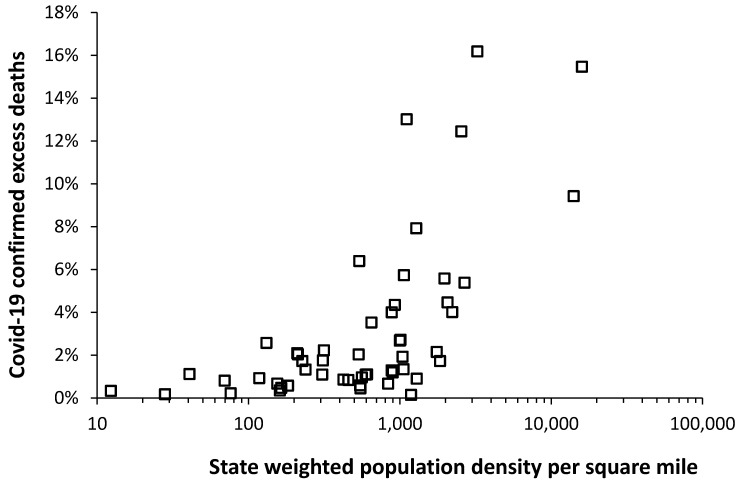
Covid-19 confirmed “excess” deaths (as at 9th June 2020) relative to 2018 as the base year versus county-level weighted state population density.

**Figure 4 ijerph-17-05210-f004:**
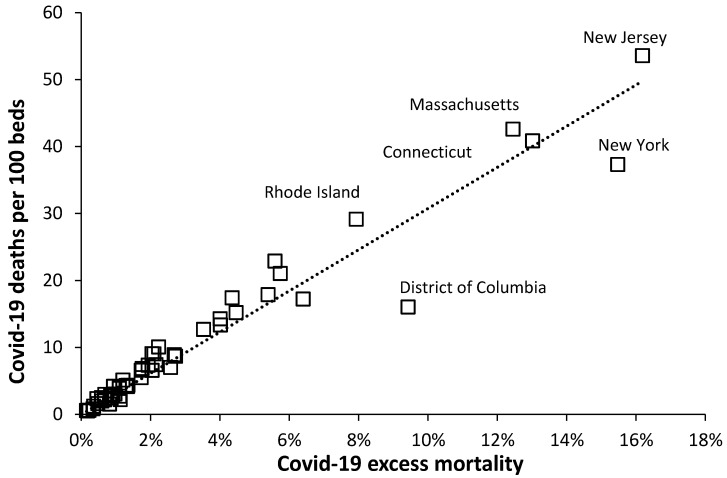
Covid-19 confirmed deaths per total hospital beds versus Covid-19 excess mortality relative to 2018 as the base year.

**Table 1 ijerph-17-05210-t001:** U.S. states with the least number of beds and nearest International equivalent, all adjusted to 8.69 deaths per 1000 population.

State	Population	Deaths	Adjusted Beds per 1000 Deaths	Nearest Equivalent
Oregon	4,181,886	36,166	191.6	Burundi
Washington	7,523,869	56,908	212.6	Costa Rica
Idaho	1,750,536	14,243	227.4	Botswana
New Hampshire	1,353,465	12,816	229.9	Botswana
Vermont	624,358	5904	230.1	Botswana
New Mexico	2,092,741	19,075	236.5	Mexico
Maryland	6,035,802	50,308	239.9	Mexico
Wisconsin	5,807,406	53,696	240.3	Guyana
Hawaii	1,420,593	11,401	249.3	Dominican Republic
Arizona	7,158,024	59,577	251.4	Ecuador
Maine	1,339,057	14,680	252.2	Algeria
Colorado	5,691,287	38,517	254.5	Algeria
Nevada	3,027,341	24,760	256.4	Thailand
Delaware	965,479	9421	262.5	Comoros
Utah	3,153,550	18,436	264.7	Peru
South Carolina	5,084,156	50,572	277.8	Fiji
Virginia	8,501,286	69,360	282.7	South Africa
Rhode Island	1,058,287	10,189	284.1	Albania
California	39,461,588	269,094	286.5	Albania
North Carolina	10,381,615	93,844	286.7	Albania
Ohio	11,676,341	124,545	286.7	Albania
Michigan	9,984,072	98,822	288.2	Iran
Massachusetts	6,882,635	59,054	290.8	Iran
Georgia	10,511,131	85,149	292.1	Andorra
Indiana	6,695,497	65,646	292.5	Trinidad and Tobago
New Jersey	8,886,025	76,002	300.2	Vanuatu
Illinois	12,723,071	109,904	300.7	Vanuatu
Texas	28,628,666	202,025	302.2	Vanuatu
Minnesota	5,606,249	44,737	302.7	Vanuatu
**U.S. average**	**324,883,210**	**2,816,380**	**300.6**	**Vanuatu**

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
