# Peer review of "Would the United States Have Had Too Few Beds for Universal Emergency Care in the Event of a More Widespread Covid-19 Epidemic?"

_ijerph, 2020, doi:10.3390/ijerph17145210_

Round 1

Reviewer 1 Report

It’s a highly mathematical analysis of the COVID-19 epidemic in relationship with the availability of hospital beds. Figure 2 shows an interesting relationship between the population density and the number of excess deaths in different states of America. Figure 3 shows the relationship between the number of deaths per 100 beds and the percentage of excess mortality.

I suggest changing the text in figure 3 and to abandon the percentage. It is just the number of deaths per 100 beds.

The weakest part of the article is paragraph 3.5.2. with the comparison of countries in the world. The way countries are dealing with this epidemic vary widely between countries. Probably a lot of statistical figures aren’t reliable. For example, in the Netherlands the mortality of Covid is underestimated for about 40% because a lack testing materials in the beginning of the epidemic. My suggestion would be to leave out this paragraph and figure 4, because it doesn’t contribute to the quality of the article the conclusion.

Author Response

This is a valid point. Section and figure removed as requested.

See the attachment.

Reviewer 2 Report

Journal: International Journal of Environmental Research and Public Health

Manuscript ID: ijerph-852649

Please see my comments on “Would the US healthcare system have had too few beds for universal emergency care in the event of a more widespread Covid-19 epidemic?”

Comments:

The paper aims to investigate the number of beds needed for an emergency in the event of Covid-19.  It feels more editorial at times than an objective measure of statistics.  A great deal of assumptions are made without supporting data.  The article discusses medical resources and high co-payment imposed by insurers, but does not, for the most part, utilize health insurance/etc. data to compare to this study’s data – What is the proportion of people who are insured in Oregon/New York/throughout the US and how does that compare with your data?  The basic premise of the article is sound by evaluating the number of beds via a new metric compared to more traditional ways.  It would have been beneficial to see how your metric compared to the traditional methods like age-based forecasting for determining hospital beds, utilizing the same dataset.  I believe you are attempting to make several points: 1. Insurance/lack of universal care, 2. Improving the metrics to better identify the need hospital beds, and 3. How Covid-19 was handled in US for bed capacity;  instead of adequately focusing on each point, with sufficient data and statistics, you briefly touched each subject.   Potentially this is due to a lack of available data. 

The introduction describes the need for the new metric, however it was slightly confusing as to the main point of the article with references to insurance.  The methods seem lacking in details, while the results seem to combine the methods, results and discussion to the point I felt the discussion was repetitive. While the new metric may be useful at determining necessary number of beds, I felt there were too many different focal points with little supporting data. 

Specific Comments:

Page/Line:

Title:  The ‘US healthcare system’ in the title implies there is a universal system for the US, which is not accurate.  I would remove the word system from the title.

1/6 – Superscript ‘h’

1/21 – ‘surprise’ is not a scientific/measurable description

1/22 – ‘world’s richest country’ according to what metric and source?  Language is editorial style verses simply stating facts, here and throughout the text.

1/25 – Why use Bing.com?  Where do they get their data from? Why not use https://www.cdc.gov/covid-data-tracker/#cases?

1/26 – You raise the question of insured patients but do not follow through with the data and investigate with further comparisons; perhaps the issue of insurance should be evaluated more deeply or removed.

1/28 – US States needs lowercase ‘s’ or change to states of the US; consider rewording – throughout the paper and figures

1/41 – A comparison of your method verses the traditional age-based hospital projection would be useful.  Information about number of hospital beds/100,000 in each city or state in US or various countries would be helpful.

1/42 – Should be added to previous paragraph

2/50 – Purpose of the sentence is unclear. 

2/55 – The article stated n=93? 

2/60 – Sentence should be added to previous paragraph.

2/73 – I would be curious to know what is the optimal bed capacity to not be wasteful/minimize spending for the hospitals and healthcare system, and yet still be prepared in extreme situations. 

2/85 – It seems like you try to present the data and how you generated it in the results, instead of stating in the methods how you were going to generate the data.  There is a lack of descriptive methods for calculations.  Who are you considering the developed and less developed countries?  What data was adjusted and why?

3/102 – Consider adding cumulative US like Canada/UK and adding (ln) to x-axis title.

5/143 – The Covid deaths are only for those who died from Covid and does not count the thousands that die on a daily basis.  I would think a true excess Covid death more like the CDC version of excess deaths: https://www.cdc.gov/nchs/nvss/vsrr/covid19/excess_deaths.htm#dashboard.

Also, did you use the deaths for 6 months from 2018, since you only have 6 months of data for Covid deaths? 

6/161 – I think a major argument missing is how nursing homes fit into this equation? Up to 80% of Covid deaths in some states are individuals living in nursing homes.  It is likely that if they are in nursing homes they do not go to hospital but are cared for in place, and therefore never require a hospital bed.  https://www.nytimes.com/interactive/2020/us/coronavirus-nursing-homes.html

6/167 – ‘Catastrophic bed capacity overload’ seems like an overstatement given all the different factors going into Covid-19 such as bed capacity, population density, disease spread, handling of the disease by the government, adherence to rules, etc. The same could be said for New Zealand as far as a catastrophic bed overload, however, seemingly their approach to the disease averted the high demand for beds.

7/179 – It raises the question of how could we/why don’t we include weighted population density into the determination of necessary beds (along with beds per death and death per 1000)?

7/191 – I don’t recall data regarding health insurance or healthcare industry with Table 1 and Figure 1.  It seems like an overstatement and oversimplification.

8/202 – Earlier in the paragraph you acknowledge you don’t know how many beds are in the US but then later say there must be barely enough.  I believe a clear definition of hospital/total/acute/staffed beds would be necessary to make any definitive statements, which you acknowledge is one of your limitations.

8/205 –There was a great concern if the US could not slow the spread of Covid that there would be a lack of hospital beds, however, I have not seen any data that demonstrate this is/was a significant problem for most states (with New York and California as possible exceptions). 

8/214 – While this argument is true, it does not add anything specific to the calculation of the number of beds needed or the lack of beds available. 

8/234 – Again overstatement, by saying ‘spared from a medical disaster’. 

9/243 - Figure 3 does not show weighted population density for each State

9/265 - Another issue that was not mentioned or accounted for is length of stay for hospitalizations which will impact the number of hospital beds available; Hospital stays have dramatically been reduced from the onset of Covid to current practice allowing for more available beds.

9/267 – While your argument may be correct, you did not provide any data or evidence ‘that the medical resources to the insured, along with high levels of co-payment imposed by insurers to discourage use of those resources has left the US with very low levels of hospital beds in relation to other developed countries.’  There was little data/discussion regarding who was insured and how that correlated with number of beds or medical resources in each state.

9/292 – Again, while it may be correct, the wording is not scientific; Reword ‘caught on the horns’.

10/all – Punctuation throughout references not consistent.

Author Response

Thank you for your detailed comments. I have attempted to address all these points. New sections have been added (in red) and sentences revised (in red).

The US healthcare 'system' is complex and insurance per se does not necessarily imply access to health care. Issues of rural poverty are introduced. However, the aim is not to give a comprehensive overview but just to demonstrate too few beds for whatever reason.

A chart showing how beds per death and beds per population has been introduced in a new section.

See the attachment.

Reviewer 3 Report

  1. In the Abstract [p. 1] it is said about “low population density” as a conclusion foundation. I recommend to re-write this sentence to following form “relatively low population density”, because the “low” meaning strongly depends on analysed country and other contexts.

  1. Lines 38-41 highlight a gap of knowledge and indicate information sources concerning current limitations in the literature sources. In my opinion it is a great chance to strengthen the practical value of the paper by additional, short description of the limitations (then references to the limitations should be considered in the discussion chapter).

  1. In line 46 a lone bracket is to delete.

  1. The Introduction poorly corresponds with the Abstract, especially due to background and methods information. I strongly recommend to modify the Introduction and, consequently a) to directly formulate knowledge gap to be matched by Author; b) to name the main objective of the research (corresponding with the gap); c) to make a clear line between current methods and the new one; d) to move detailed information about the new method to the second chapter (the methodological one).

  1. In the Materials and Methods chapter there is no description of calculation mechanisms. The chapter does not inform how to repeat the research method. From this reason the methodology dimension is not sound. So, I recommend to consider detail information about the calculation mechanism (lines 44-59 could be helpful) in order reflecting a structure of the Results chapter. Division to subchapters respecting the Results chapter structure is highly desirable.

  1. In 2.2. (line 85) it is mentioned about data manipulation and charts prepared using MsExcel. To make the methodology clear, I recommend to add the charts or just a chart with joint data from the charts as an annex to the manuscript.

  1. Line 246 – why “Cities” word is wrote down using capital letter? What difference is between city and town in analysed context?

  1. In my opinion, description of limitations (subchapter 4.7.) should consider an analysis of beds in COVID-19 solitary premises (hostels, churches, gyms and other buildings adopted to isolation requirements) as only such context could give a real situational picture.

  1. Generally, chapter 4 idea (so the relevant structure as well) is not obvious and makes some interpretation problems. Reader does not know why exactly such issues are discussed in presented order. I think that clear presentation of the main research objective in the Instruction and close reference to the Materials and Methods should increase the research soundness.

  1. The Conclusions chapter contains quite obvious information that could be formulated (even) without the research. So, it is required to add more information directly connected to the research results.

  1. Furthermore, if the Conclusions chapter takes into consideration information about further research directions (basing on predefined limitations, maybe), it will increase the practical value of the research. To do this, I recommend to add also answer to a question – how the results can be used/implemented in practice.

Author Response

Thank you for your comprehensive suggestions which I have attempted to address. New sections have been added (in red) along with additional references (in red) and additional explanation of the relative importance of issues such as percentage rural population and other factors (in red).

Issues around capital letters, etc have been addressed.

See the attachment.

Round 2

Reviewer 2 Report

Journal: International Journal of Environmental Research and Public Health

Manuscript ID: ijerph-852649

Please see my comments on “Would the United States have had too few beds for universal emergency care in the event of a more widespread Covid-19 epidemic?

Response:

Throughout: There are many 1 or 2 word sentence paragraphs that can be combined into more comprehensive paragraphs.

 3/92 – “3.1” Wrong section designation.

5/154 – The following belongs in the discussion, as it is not information from your results:  “As can be seen Oregon, a largely agricultural State with lower paid farm workers [34,35]”

6/163 – Incorrect font size and italics? : “3.5.Countries near to the less developed countries average are Colombia, Ecuador, and Algeria”

2/85 and 12/452 – “CDC” in sentence, citation author and website; Also, should be defined once (Centers for Disease Control and Prevention)

7/205 – D.C. already defined

7/210 – Consider moving “4.1. Healthcare in the US” to after Population Density and before Limitations.  It is not the highlight of your research.

10/349 – Length of stay should not be an issue raised during conclusions (possibly move to Limitations).   Also, when previously asked about length of stay and unavailable hospital beds, I was asking if there is any current literature specifically regarding these issues and Covid available in the US that can be added.

References were not fixed. Period after article title? Comma after journal name? etc.

Author Response

Response:

Throughout: There are many 1 or 2 word sentence paragraphs that can be combined into more comprehensive paragraphs.

As a communication tool I like to use short paragraphs, especially for the benefit of managers. However, I have tried to concatenate.

  3/92 – “3.1” Wrong section designation.

 Section 3.1 now moved into methods section. Results sections renumbered

5/154 – The following belongs in the discussion, as it is not information from your results:  “As can be seen Oregon, a largely agricultural State with lower paid farm workers [34,35]”

 Thank you, duly moved.

6/163 – Incorrect font size and italics? : “3.5.Countries near to the less developed countries average are Colombia, Ecuador, and Algeria”

 Everything looks correct

2/85 and 12/452 – “CDC” in sentence, citation author and website; Also, should be defined once (Centers for Disease Control and Prevention)

 Changed

7/205 – D.C. already defined

 Changed

7/210 – Consider moving “4.1. Healthcare in the US” to after Population Density and before Limitations.  It is not the highlight of your research.

 This section has been placed at the start of the discussion to highlight how complex ‘healthcare’ is in the US. Section 4.7 has been added to attempt to further elaborate on this complexity in relation to bed numbers.

10/349 – Length of stay should not be an issue raised during conclusions (possibly move to Limitations).   Also, when previously asked about length of stay and unavailable hospital beds, I was asking if there is any current literature specifically regarding these issues and Covid available in the US that can be added.

 This is where it all gest a bit confusing. In the UK the issue was considered so politically sensitive that statistics regarding CCU and other bed numbers were ‘suspended’, meaning, no one can do any rational calculations of what went on. Dumping of patients into nursing homes after initial; acute treatment to keep free beds also occurred in New York and maybe elsewhere. The same happened in the UK with disastrous results. Who knows what bed pressures actually occurred?

I didn’t really want to go into such murky details as it is secondary to the main aim regarding using a new method to raise questions about comparative bed numbers. A concise summary has been added to the text.

References were not fixed. Period after article title? Comma after journal name? etc.

Done my best to correct, alas old guy using reference cards, etc.

Reviewer 3 Report

All my remarks have been taken into consideration. The manuscript has been evaluated to increase the scientific soundness and overall research presentation quality.

Author Response

No further actions required.